# Optimization of Solid-State Fermentation Extraction of *Inonotus hispidus* Fruiting Body Melanin

**DOI:** 10.3390/foods10122893

**Published:** 2021-11-23

**Authors:** Fengpei Zhang, Fanzheng Xue, Hui Xu, Yuan Yuan, Xiaoping Wu, Junli Zhang, Junsheng Fu

**Affiliations:** 1College of Life Science, Fujian Agriculture and Forestry University, Fuzhou 350002, China; zhangfengpei0531@163.com (F.Z.); xfzzzzz719@163.com (F.X.); xuhui189500@163.com (H.X.); 18388376609@163.com (Y.Y.); fjwxp@126.com (X.W.); 2Mycological Research Center, College of Life Sciences, Fujian Agriculture and Forestry University, Fuzhou 350002, China; 3Tibet Academy of Agricultural and Animal Husbandry Sciences, Lhasa 850000, China; zjunli2021@163.com

**Keywords:** melanin, optimization, structure, *Inonotus hispidus*

## Abstract

Melanin has good nutritional and medicinal value; however, its extraction rate is extremely low. This study explored the edible and medicinal fungus *Inonotus hispidus* fruiting body melanin (IHFM) extraction process and solid-state fermentation conditions. The results showed that the best way to extract IHFM is the compound enzymatic method, with complex enzyme 26.63 mg/g, liquid material ratio 5:1, enzymatic hydrolysis 80 min, pH 4.61, and enzymolysis temperature at 36.07 °C. The yield of IHFM was 23.73 ± 0.57%, which was equivalent to 1.27 times before optimization. The best solid medium formula was normal pH, rice 20 g per cultivation bottle, maltose 22 g/L, beef extract 4.4 g/L, carbon-nitrogen ratio 5:1, and liquid-to-material ratio 1.1:1, where the IHFM yield was 31.80 ± 1.34%, which was equivalent to 1.7 times that before optimization. In summary, solid-state fermentation and extraction optimization greatly improved the yield of melanin, provided a reference to produce melanin, and laid a foundation for the development and utilization of melanin.

## 1. Introduction

Melanin is an extremely widely distributed class of pH biological pigments, generally dark brown to black in color, with good nutritional and medicinal values, widely used in several fields such as food additives and health care [1,2]. It is a macromolecular polymer formed by cyclic subunits, such as indole, phenol, and hydroxynaphthol, which are commonly found in animals, plants, and protozoa [3,4]. The reports of melanin in edible fungi are generally concentrated in *Auricularia auricula* and *Auricularia polytricha* while other edible fungi have low melanin content or almost no melanin. As a natural active ingredient, melanin has wide medicinal value and biological activity [5,6], which can convert 99.9% of the energy from sunlight into thermal energy, effectively reducing the probability of skin cancer [7]. Shi et al. found that *lachnum* YM226 melanin not only promoted apoptosis of hepatocellular carcinoma cells, but also had a good antihyperlipidemic effect, significantly reducing low-density lipoprotein cholesterol (LDL-C) and increasing high-density lipoprotein cholesterol (HDL-C) in mice, suggesting that natural melanin is a potential anticancer and lipid-lowering active substance [8,9].

Wild *Inonotus hispidus* normally grow to golden yellow fruiting body during summer and autumn. After late autumn, the fruiting body gradually ages, the color changes from light brown to dark brown, and eventually turns to black [7]. Current studies on *Inonotus hispidus* have focused on active ingredients such as polysaccharides, polyphenols, and flavonoids [10], with less research on melanin. Our team previously extracted *Inonotus hispidus* fruiting body melanin (IHFM) from the culture matrix and found it had good antioxidant activity against DPPH radicals and hydroxyl radicals [11]. However, due to the low extraction rate and the limitations of growth cycle and environmental factors, it is not conducive to the development and application of the IHFM, so it is necessary to explore the solid fermentation conditions and extraction process with higher melanin production rate.

The traditional extraction of melanin uses alkaline extraction and the acid-precipitation method, but the extraction rate is low [12,13]. Relatively speaking, the enzymatic extraction is simple and efficient, which can rapidly disrupt the cell wall and promote the leaching of active substances inside the cells, which has specificity, high efficiency, and mild reaction conditions [11]. The compound enzyme synergistic extraction process refers to the combination of two or more enzymes, to disrupt multiple structures of the cells, thus achieving efficient extraction of active substances [14]. Our team used cellulase to extract melanin in the early stage, which improved the IHFM extraction rate [11]. To obtain the best compound enzyme synergistic extraction process for IHFM, we further use cellulase, pectinase, and papain to perform single-factor extraction experiments and response surface optimization. Besides this, we used orthogonal test to optimize the fermentation conditions of *Inonotus hispidus* rice medium so as to achieve high melanin production from *Inonotus hispidus* fruiting body.

## 2. Materials and Methods

### 2.1. Materials

The materials are as follows: *Inonotus hispidus* strain MS-5 (isolated by our laboratory from the cysts of *Inonotus hispidus* substrates growing on wild jujube trees in Zhongwei, Ningxia, China (NCBI accession number: MF183947)); Rice (Fujian RT-Mart, Fujian, China); Dialysis bags (Beijing Solarbio Technology Co., Beijing, China). All other analytical grade chemical reagents were from the Sinopharm Co. (Shanghai, China).

### 2.2. Single Factor Extraction Optimization and Response Surface Optimization of IHFM

*The synergistic extraction of IHFM with compound enzyme process* Based on the method of Hou [11], with slight modifications. The collected *Inonotus hispidus* fruiting bodies were dried at 60 °C to constant weight, crushed, and passed through 80 mesh sieve fully. Then the powder was weighed to 0.5 g accurately and reacted in the enzymatic solution with certain enzyme ratio, enzyme addition, enzymatic pH, liquid-to-material ratio and enzymatic temperature at a certain temperature for a period of time. After the enzymatic process was finished, we extracted the precipitate by centrifugation for 5 min at 10,000 rpm. Among them, cellulase was dissolved in sodium acetate buffer pH 4.6, and pectinase and papain were dissolved in water. The precipitate obtained after enzymatic hydrolysis was dissolved in 1.5 mol/L NaOH according to the material-to-liquid ratio of 1:30. After fully reacting for 1 h, the supernatant was collected by centrifugation at 1000 r/min for 5 min, and concentrated hydrochloric acid was used to obtain the supernatant. The pH of the solution was adjusted to 1.5–2.0, placed in a water bath at 80 °C for 10 h, and the precipitate was collected by centrifugation as a crude melanin product. We then repeated the previous step, rinsed the precipitate repeatedly with distilled water until it was neutral, and then rinsed with organic reagents chloroform, dichloromethane, ethyl acetate, absolute ethanol, 75% ethanol and distilled water in turn, centrifuge at 10,000 r/min for 5 min and obtain the precipitate, and freeze-dried to obtain a pure soluble melanin product. The calculation of IHFM yield is shown in the formula:Melanin yield (%) = crude melanin mass/fruit body powder mass × 100(1)

*Single factor extraction conditions design* The IHFM extraction rate was used as an indicator to investigate the ratio of compound enzyme (0:0:0, 1:0:1, 0:1:1, 1:1:0, 1:1:1, 2:1:0, 2:0:1, 1:2:0, 1:0:2, 0:1:2, 0:2:1, 1:3:0, 1:0:3, 3:0:1, 3:1:0, 0:3:1, 0:1:3, 1:1:2, 1:2:1, 2:1:1), enzyme addition (0, 5, 10, 15, 20, 25, 30 mg/g), liquid-to-material ratio (5:1, 10:1, 15:1, 20:1, 25:1, 30:1, 35:1), enzymatic pH (3, 3.5, 4, 4.5, 5, 5.5, 6), enzymatic temperature (30, 35, 40, 45, 50, 55 and 60 °C), and enzymatic time (20, 40, 60, 80, 100, 120, 140 min) on the extraction rate of IHFM.

*Box–Behnken design* Based on the Box–Behnken central combination principle, we used Design expert 11.1.0.1 software to conduct response surface test on single factor test results to optimize the IHFM compound enzyme extraction process, and the specific response surface test design is shown in Table 1.

### 2.3. Single-Factor Optimization and Orthogonal Experiment of Solid-State Fermentation Conditions for IHFM

*Alkaline extraction of melanin* Based on the method of Hou Ruolin et al., with slight modifications [11]. We ground and sieved the fruiting bodies of *Inonotus hispidus*, obtaining the crude melanin according to the alkaline extraction method. The melanin precipitate was washed repeatedly with distilled water to neutrality, and then we used the organic reagents chloroform, dichloromethane, ethyl acetate, absolute ethanol, 75% ethanol and distilled water, centrifuge at 10,000 rpm for 5 min to obtain a precipitate, and obtained pure insoluble melanin by freeze-drying. The precipitation in the previous step was first reconstituted with 0.2 mol/L NaOH, and then adjusted to normal pH with 0.1 mol/L HCl, dialyzed in running water for 48 h, and freeze-dried to obtain a pure soluble melanin product. The calculation method of melanin yield refers to Formula (1).

*Preparation of melanin by solid-state fermentation* In order to explore the rice fermentation medium that can promote the large amount of melanin produced by *Inonotus hispidus*, the fermentation broth of *Inonotus hispidus* was diluted 10 times to prepare a seed culture solution [15]. According to the liquid-to-material ratio of 1.1:1, 6 mL of *Inonotus hispidus* culture solution was inoculated into the optimized rice culture medium, and the cultivation bottle was placed in a constant temperature incubator at 25 °C for fermentation and culture for 90 days. During the growth of the hyphae, the culture should be protected from light [16]. When the hyphae were overgrown with the rice medium, they were cultivated alternately in light and dark for 12 h every day, with a light intensity of 850–1050 lx and a humidity of 85–90%. After the completion of the cultivation, the fruit bodies of *Inonotus hispidus* were collected and dried in an oven at 60 °C, and the melanin was extracted.

*Single factor fermentation conditions design* The extraction yield of IHFM was used as an indicator to explore the carbon sources (maltose, fructose, mannitol, starch, corn flour, glucose, sucrose, xylose, and lactose) and nitrogen sources (ammonium nitrate, beef Extract, ammonium tartrate, tryptone, potassium nitrate, urea, and yeast extract powder), carbon-to-nitrogen ratio (5:1, 10:1, 20:1, 30:1, 40:1, 50:1, 60:1 and 70:1), liquid-to-material ratio (1:1, 1.1:1, 1.2:1, 1.3:1, 1.4:1 and 1.5:1), pH (5, 5.5, 6, 6.5, 7, 7.5 and 8), and other factors (tyrosine 0.5, 1, 1.5, 2, 2.5 and 3 g/L) on the melanin production of *Inonotus hispidus* in rice cultivation.

*Orthogonal test* Based on the single factor experiment, the experiment further carried out orthogonal optimization of carbon source, nitrogen source, pH, and liquid-to-material ratio, and set three replicates for each group. See Table 2 for details.

### 2.4. Determination of Infrared Absorption Spectroscopy

Using the potassium bromide tableting method, 1~2 mg purified IHEM powder sample and 200 mg KBr pure powder were mixed and pressed into a transparent sheet. For each sample, 32 scans of the infrared region between 4000 and 400 cm^−1^ at a resolution of 4 cm^−1^ were recorded in triplicates and averaged [17].

### 2.5. Scanning by Electron Microscope

A quantity of 10 mg of the soluble IHFM was sprayed with gold for 50 s to enhance the conductivity of the sample, and then the sample was directly glued to the conductive glue for testing, and the surface morphology of the melanin was observed and photographed [18].

### 2.6. Statistical Analysis

All data was calculated with SPSS 25.0 statistical software for the experimental data. The measurement data (*x* ± *s*) is the mean ± standard deviation. The *t* test was used to analyze the significance of the two groups, *p* < 0.05 indicates a significant difference, and *p* < 0.01 the difference is extremely significant.

## 3. Results

### 3.1. Single Factor Analysis of IHFM Compound Enzyme Extraction Conditions

*Enzyme ratio* Under the conditions of enzyme addition 20 mg/g, liquid-to-material ratio 20:1, normal pH, enzymolysis temperature 40 °C, enzymolysis time 60 min, we explore the enzyme ratio of cellulase, pectinase, and papain at 0:0:0, 1:0:1, 0:1:1, 1:1:0, 1:1:1, 2:1:0, 2:0:1, 1:2:0, 1:0: 2, 0:1:2, 0:2:1, 1:3:0, 1:0:3, 3:0:1, 3:1:0, 0:3:1, 0:1:3, 1:1:2, 1:2:1, 2:1:1, and their respective effect on the extraction yield of IHFM. Compared with the blank group, the compound enzyme ratio of 2:1:0, 2:0:1, 1:2:0, 1:0:2, 0:1:2, 0:2:1, and 1:0:3 significantly increase the yield of IHFM (Figure 1A). Among them, the yield of IHFM is the highest when the enzyme ratio of the composite enzyme is 0:1:2, so it is more appropriate to choose the enzyme ratio 0:1:2 to extract IHFM.

*Enzyme addition* Using the enzyme ratio of 0:1:2, liquid-to-material ratio 20:1, the pH is normal, the enzymatic hydrolysis temperature at 40 °C, and the enzymatic hydrolysis time of 60 min, we explore the effect of enzyme dosage 0, 5, 10, 15, 20, 25, 30 mg/g on the extraction rate of IHFM. When the enzyme dosage is 25 mg/g, the yield of IHFM is the highest, and then it decreases (Figure 1B). Therefore, it is more appropriate to use 25 mg/g enzyme addition amount to extract IHFM.

*Liquid-to-material ratio* Using the enzyme ratio of 0:1:2, enzyme addition amount of 25 mg/g, the pH is normal, the enzymatic hydrolysis temperature at 40 °C, and the enzymatic hydrolysis time of 60 min, we explore effect of the liquid-to-material ratio 5:1, 10:1, 15:1, 20:1, 25:1, 30:1, and 35:1 on the extraction yield of IHFM. The yield of IHFM decreases significantly with the increase of the liquid-to-material ratio (Figure 1C). When the liquid-to-material ratio is 5:1, the IHFM yield is the highest. Therefore, it is more appropriate to pick a liquid-to-material ratio of 5:1 to extract IHFM.

*Enzymatic pH* Using the enzyme ratio of 0:1:2, enzyme addition amount of 25 mg/g, the liquid-to-material ratio of 5:1, the enzymatic hydrolysis temperature at 40 °C, and the enzymatic hydrolysis time of 60 min, we explore the influence of enzymatic hydrolysis pH 3, 3.5, 4, 4.5, 5, 5.5 and 6 on the extraction yield of IHFM. When the pH of enzymatic hydrolysis is 4.5, the yield of IHFM is the highest, and then began to decline (Figure 1D). Therefore, it is more appropriate to choose pH 4.5 to extract IHFM.

*Enzymatic hydrolysis temperature* Using the enzyme ratio of 0:1:2, amount of enzyme added of 25 mg/g, the liquid-to-material ratio of 5:1, the pH of the enzymatic hydrolysis is 4.5, and the enzymatic hydrolysis time of 60 min, we explore the influence of enzymatic temperature of 30, 35, 40, 45, 50, 55, and 60 °C on the extraction yield of IHFM. As the enzymatic hydrolysis temperature increases, the yield of IHFM first increases and then decreases (Figure 1E). When the enzymatic hydrolysis temperature is 35 °C, the yield of IHFM is the highest. Therefore, it is more appropriate to select enzymatic hydrolysis temperature at 35 °C to continue the extraction of IHFM.

*Enzymatic hydrolysis time* Using the enzyme ratio of 0:1:2, amount of enzyme added of 25 mg, the liquid-to-material ratio of 5:1, the pH of the enzymatic hydrolysis is 4.5, and the enzymatic temperature at 35 °C, we explore the influence of enzymatic hydrolysis time of 20, 40, 60, 80, 100, 120, and 140 min on the extraction yield of IHFM. The yield of IHFM is the highest when the enzymatic hydrolysis time is 80 min (Figure 1F). Therefore, it is more appropriate to choose 80 min as enzymatic hydrolysis time to continue the extraction of IHFM.

### 3.2. Response Surface Analysis

To further improve the extraction yield of IHFM, the experiment takes IHFM extraction yield (*Y*) as the response value, selects A (enzyme addition), B (enzymatic hydrolysis pH) and C (enzymatic hydrolysis temperature) as independent variables, and designs three factors for the horizontal response surface analysis test (Table 3). By analyzing the IHFM yield of each optimized combination, the regression equation between A, B, C 3 factors and Y was: *Y* = 23.48 + 0.3917A + 0.3917B + 0.3C + 0.05AB − 0.1333BC − 0.615A^2^ − 0.8483B^2^ − 0.6317C^2^.(2)

Through the regression model analysis of variance and significance, it was found that the significance of the model is *p* < 0.0001, indicating that the model was extremely significant (Table 4). The equation correlation coefficient *R*^2^ = 0.9772 indicated that the model had a good degree of fit and it could effectively reflect the relationship between the three single factors and the response value *Y*. At the same time, the lack of fit term p of the regression equation was not significant, indicating that the regression equation was more consistent with the actual value, and the error was relatively small. Therefore, the above regression equation could be used to analyze and predict the optimal compound enzyme extraction conditions for IHFM.

The linear coefficients A, B, and C of the regression equation and the quadratic coefficients A^2^, C^2^ and B^2^ have very significant effects (*p* < 0.01). The response surface is drawn based on the analysis of the above regression equation to determine the influence of each factor on the extraction yield of IHFM (Figure 2). The analysis results found that the optimal compound enzyme extraction conditions predicted by the model were: compound enzyme addition amount 26.63 mg/g, enzymatic hydrolysis pH 4.61, and enzymatic hydrolysis temperature 36.07 °C. Under these conditions, the IHFM yield was expected to reach 23.62%.

### 3.3. Single Factor Analysis of IHFM Fermentation Conditions

*Carbon source* We used the rice fermentation medium mentioned in Section 2.3 as the basic medium, and the blank control medium without carbon source. The carbon sources were replaced with maltose, fructose, mannitol, starch, corn flour, glucose, sucrose, xylose, and lactose. They were fermented and cultured in an incubator for 90 days to explore the best carbon source that can effectively promote the production of IHFM. The fruiting production of *Inonotus hispidus* was different in each group (Figure 3A). When maltose was used, the extraction yield of IHFM was the highest, so we selected maltose as the best carbon source for further optimization.

*Nitrogen source* We used the rice fermentation medium without added nitrogen source as the blank control. The nitrogen sources were ammonium nitrate, beef extract, ammonium tartrate, tryptone, potassium nitrate, urea, and yeast extract powder. These were fermented in an incubator for 90 days to explore the best nitrogen source. Under the condition of beef extract, the yield of IHFM was the highest (Figure 3B). Therefore, we chose beef extract as the nitrogen source of the rice fermentation medium to continue optimization.

*Carbon-to-nitrogen ratio* The carbon-to-nitrogen ratios were 5:1, 10:1, 20:1, 30:1, 40:1, 50:1, 60:1, and 70:1 under maltose and beef extract. The IHFM content decreases with the increase of the carbon-to-nitrogen ratio. When the carbon-to-nitrogen ratio was 5:1, the IHFM yield was the highest (Figure 3C). Therefore, it was selected as the carbon-to-nitrogen ratio of the rice fermentation medium to continue to optimize More appropriate.

*Liquid-to-material ratio* The liquid-to-material ratios were 1:1, 1.1:1, 1.2:1, 1.3:1, 1.4:1, and 1.5:1. The yield of IHFM was the highest when the liquid-to-material ratio was 1.1:1 (Figure 3D). Therefore, we chose the liquid-to-material ratio of 1.1:1 to continue optimization.

*pH* With normal pH as the control (pH = 6.7), the medium pH was adjusted to 5, 5.5, 6, 6.5, 7, 7.5, and 8. Adjusting the pH of the medium will affect the yield of IHFM and fruiting to varying degrees. The IHFM content was the highest under the normal pH; thus, we chose normal pH to continue optimization (Figure 3E).

*Other factors* We added tyrosine 0.5, 1, 1.5, 2, 2.5, and 3 g/L. As shown in Figure 3F, when 2.5 g/L tyrosine was added to the medium, the melanin yield was the highest, so we chose 2.5 g/L tyrosine to continue the orthogonal optimization.

### 3.4. Orthogonal Analysis

Taking IHFM yield as an indicator, we selected the carbon source, nitrogen source, pH and liquid-to-material ratio to perform an orthogonal test of three levels and four factors.

The intuitive analysis results of the orthogonal test are shown in Table 5. The best culture conditions which can effectively promote the yield of IHFM were maltose, beef extract, normal pH, and the liquid-to-material ratio of 1.1:1. The IHFM yield was as high as 31.8 ± 1.34% under this optimized combination. Through the comparison and analysis of the range R, it was found that the range of the four factors from largest to smallest was pH, nitrogen, source, carbon source, and liquid-to-material ratio, indicating that pH was the main factor affecting the content of IHFM.

The analysis results of variance are shown in Table 6. The F values of four factors from largest to smallest were also pH, nitrogen source, carbon source, and liquid-to-material ratio, which was consistent with the results of intuitive analysis.

### 3.5. Solid Fermentation Extraction of IHFM under Compound Enzyme Method

We used the compound enzyme extraction method to extract melanin from the fruiting body. The optimized formula was cellulase, pectinase, and papain with a ratio of 0:1:2, liquid-to-material ratio 5:1, enzymatic time 80 min, pH value 4.61, the addition amount of enzyme 26.63 mg/g, and enzymatic temperature 36.07 °C. The actual yield of IHFM obtained was 23.73 ± 0.57%, which was equivalent to 1.27 times before optimization.

The optimization of solid fermentation culture could effectively improve the yield of IHFM. The optimized rice medium was 22 g/L maltose, 4.4 g/L beef extract, carbon-to-nitrogen ratio 5:1, tyrosine 2.5 g/L, 1.1:1 ratio of liquid to material, and the normal pH. The IHFM yield was as high as 31.80 ± 1.34% under these conditions, which was equivalent to 1.7 times that before optimization.

The compound enzyme could significantly improve the yield of IHFM. After optimizing extraction conditions, the extraction rate of IHFM obtained by solid fermentation optimization was as high as 33.60 ± 0.65%, which was equivalent to 1.80 times that before optimization (Figure 4).

### 3.6. Structure Analysis of Soluble IHFM

Infrared spectroscopy analysis is one of the common methods used to resolve the structure, and each group has a specific infrared absorption peak [19]. We used infrared spectrometer to determine the infrared spectrum of IHFM and found that soluble IHFM had a strong and broad characteristic absorption at 3313.39 cm^−1^, corresponding to the –NH group attached to the –OH group in the indole ring. The value 2963.11 cm^−1^ corresponded to aliphatic –CH while 1653.36 cm^−1^ corresponded to C=O stretching or aromatic C=C stretching. The peak around 1542.40 cm^−1^ was associated with –NH bending vibration, and 1407.78 cm^−1^ was associated with –CN stretching peak, suggesting an indole structure. At 611.02 cm^−1^, the aromatic ring became conjugated to the conjugated system (Figure 5A). All the above features indicated that the IR spectra of IHFM obtained by extraction and purification were consistent with the structural features of traditional melanin.

Scanning electron microscopy has the advantages of wide observation area (from centimeters to nanometers), large depth of field, high resolution, and good three-dimensional image. It could not only use secondary electron images to characterize the structure of the material, but also use backscattered electrons. The image is used to set off the composition information of materials, so it is widely used in many fields of scientific research [20]. The surface of the soluble IHFM was flake-shaped, mainly composed of irregular particles of 10–15 nm (Figure 5B).

## 4. Discussion

Since melanin is soluble in alkaline solution, it is traditionally extracted by alkali-solubilization and acid-precipitation method [21]. Yao et al. used sodium hydroxide to extract melanin from the seed coat of *Prunus armeniaca* with a yield of 4.73% [22]. Wu et al. used sodium hydroxide to extract melanin from *Auricularia Auricula* and optimized it by orthogonal test with a melanin yield of 7.19% [23]. Hou et al. used cellulase assisted ultrasonic extraction of *Auricularia Auricula* melanin, and the yield could reach 10.48% [24]. Lu et al. used both enzymatic and ultrasonic methods to extract intracellular melanin from *Lachnum*, and the melanin yield was up to 12.50%; these results suggested that enzymatic method can effectively improve the melanin extraction rate [25]. The compound enzyme extraction process is a synergistic extraction method of multiple enzymes, which achieves efficient extraction of active substances by destroying multiple intracellular structures [14]. Our research used cellulase, pectinase, and papain to extract IHFM according to certain enzyme ratios, enzyme digestion time, enzyme digestion temperature, and enzyme digestion pH, and the final extraction rate of IHFM could reach 23.73 ± 0.57%, which is equivalent to 1.27 times that before optimization.

In the early stage, our group produced melanin through liquid fermentation, then we later found a large amount of melanin in *Inonotus hispidus* cultivated in rice, and the melanin had good thermal stability and light stability [26,27]. Therefore, the experiment further optimized the extraction process and fermentation conditions of *Inonotus hispidus* to obtain a solid-state fermentation extraction method with high yield of IHFM. The soluble IHFM functional group and microstructure analysis verified the analysis of melanin structure by [11].

## 5. Conclusions

In summary, the compound enzyme significantly improved the yield of IHFM; under the process-optimized extraction conditions, the extraction rate of IHFM was significantly improved by solid fermentation optimization, which provides a reference scheme for the factory production of melanin.

## Figures and Tables

**Figure 1 foods-10-02893-f001:**
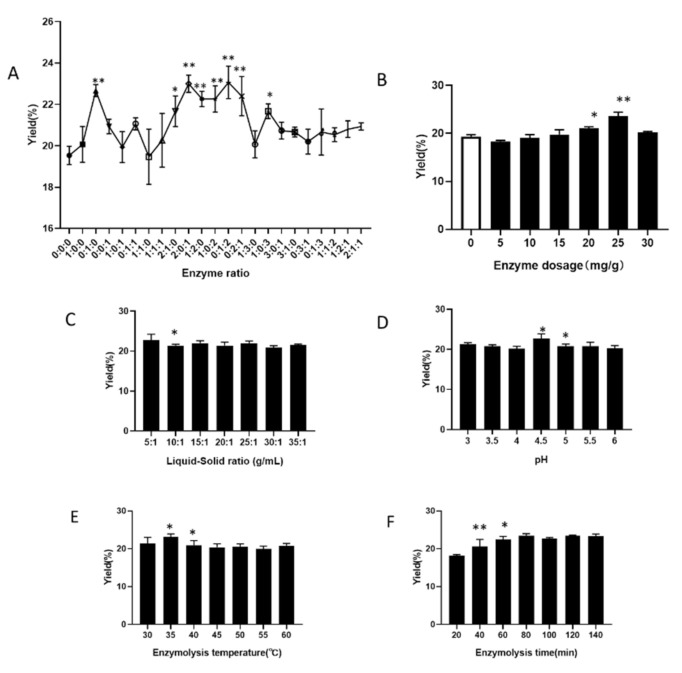
Effect of different enzymatic hydrolysis factors on the melanin yield of *Inonotus hispidus*. (**A**) Enzyme ratio; (**B**) Enzyme addition; (**C**) Liquid-to-material ratio; (**D**) Enzymatic pH; (**E**) Enzymatic hydrolysis temperature; (**F**) Enzymatic hydrolysis time. (**A**,**B**) compared with the blank group; (**C**–**F**) compared with the front group. Superscript characters indicate significant variation, * *p* < 0.05, ** *p* < 0.01.

**Figure 2 foods-10-02893-f002:**
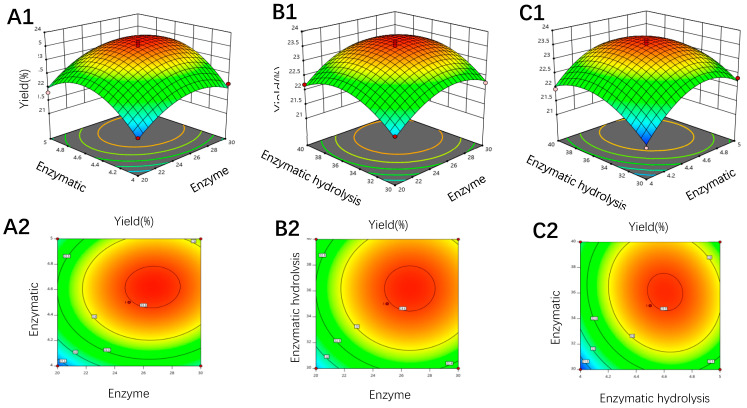
Response surface plot showing the effect of (**A1**,**A2**) enzymatic hydrolysis pH and enzyme amount; (**B1**,**B2**) enzymatic hydrolysis temperature and enzyme amount; (**C1**,**C2**) enzymatic hydrolysis temperature and enzymatic hydrolysis pH.

**Figure 3 foods-10-02893-f003:**
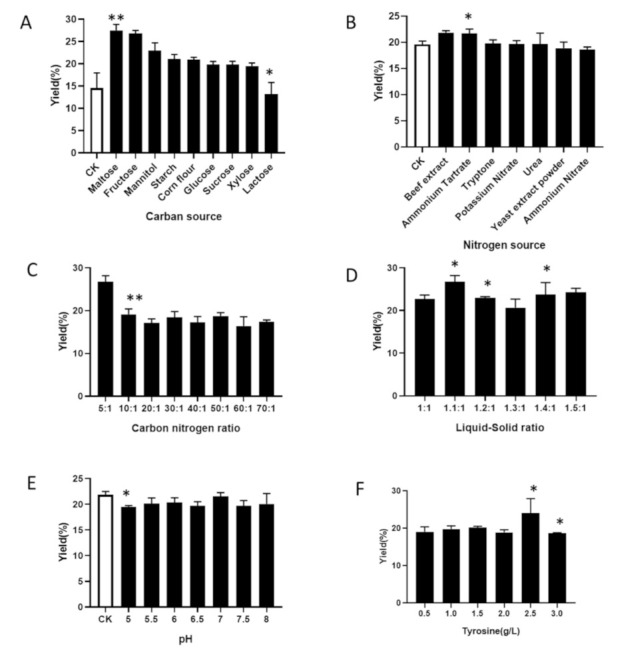
Effect of different factors on the melanin yield of *Inonotus hispidus*. (**A**) Carbon source; (**B**) Nitrogen source; (**C**) Carbon-to-nitrogen ratio; (**D**) Liquid-to-material ratio; (**E**) pH; (**F**) Tyrosine. (**A**,**B**,**E**) compared with the blank group; (**C**,**D**,**F**) compared with the front group. * *p* < 0.05, ** *p* < 0.01.

**Figure 4 foods-10-02893-f004:**
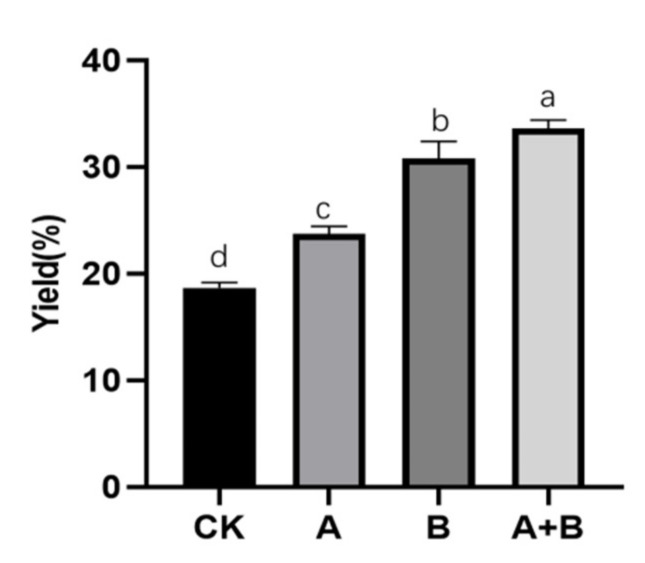
Actual yield of melanin in the compound enzyme group and the non-complex enzyme group. Group A: Compound enzyme group; Group B: Solid fermentation optimization group; Group A + B: Compound enzyme with solid fermentation optimization group. Compared with the front group. Different lowercase letters represent significant differences (*p* < 0.05, Duncan).

**Figure 5 foods-10-02893-f005:**
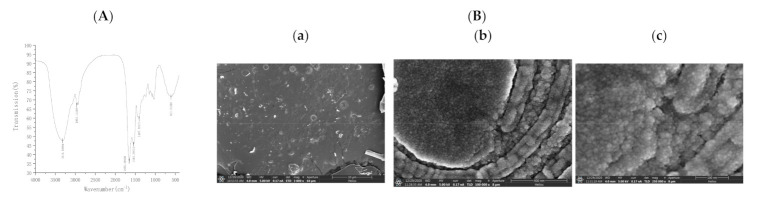
(**A**) Infrared absorption spectrum of soluble IHFM; (**B**) Scanning electron micrograph of soluble IHFM, (**a**) 10 µm; (**b**) 400 nm; (**c**) 100 nm.

**Table 1 foods-10-02893-t001:** Factors and levels of the Box–Behnken test.

Level	Factors		
	A: Enzyme Addition (mg/g)	B: Enzymolysis pH	C: Enzymolysis Temperature (°C)
−1	20	4	30
0	25	4.5	35
1	30	5	40

**Table 2 foods-10-02893-t002:** Factors and levels of the orthogonal experiment.

Level	Factors			
	Carbon Source	Nitrogen Source	pH	Liquid: Solid
1	Maltose	Beef extract	Nature	1.1:1
2	Fructose	Ammoniumtatrat	6	1.5:1
3	Mannitol	Tryptone	7	1.4:1

**Table 3 foods-10-02893-t003:** Response surface design and test results.

No.	A: Enzyme Amount (mg/g)	B: Enzymatic Hydrolysis pH	C: Enzymatic Hydrolysis Temperature (°C)	Yield (%)
1	25	4.5	35	23.33 ± 0.25
2	25	4	40	21.93 ± 0.25
3	20	4.5	30	21.60 ± 0.16
4	25	4.5	35	23.53 ± 0.57
5	20	4	35	21.33 ± 0.25
6	20	5	35	21.80 ± 0.43
7	25	4	30	21.07 ± 0.41
8	20	4.5	40	22.20 ± 0.59
9	25	5	40	22.67 ± 0.57
10	25	5	30	22.33 ± 0.25
11	30	5	35	22.80 ± 0.43
12	30	4.5	30	22.27 ± 0.52
13	25	4.5	35	23.27 ± 0.09
14	30	4.5	40	22.87 ± 0.50
15	25	4.5	35	23.60 ± 0.16
16	25	4.5	35	23.67 ± 0.50
17	30	4	35	22.13 ± 0.81

**Table 4 foods-10-02893-t004:** Variance analysis and significance tests.

Source	Sum of Squares	Df	Mean Squares	F	*p*
Model	10.28	9	1.14	33.27	<0.0001
A	1.23	1	1.23	35.76	0.0006
B	1.23	1	1.23	35.76	0.0006
C	0.7200	1	0.7200	20.98	0.0025
AB	0.0100	1	0.0100	0.2914	0.6061
AC	0.0000	1	0.0000	0.0000	1.0000
BC	0.0711	1	0.0711	2.07	0.1932
A^2^	1.59	1	1.59	46.41	0.0003
B^2^	3.03	1	3.03	88.30	<0.0001
C^2^	1.68	1	1.68	48.96	0.0002
Lack of fit	0.1211	3	0.0404	1.36	0.3758
Residual error	0.2402	7	0.0343	*R*^2^ = 0.9772	
Pure error	0.1191	4	0.0298	Adj *R*^2^ = 0.9478	
Total	10.52	16			

Note: A: Enzyme amount (mg/g); B: Enzymatic hydrolysis pH; C: Enzymatic hydrolysis temperature (°C).

**Table 5 foods-10-02893-t005:** Intuitionistic analysis of the orthogonal test results on melanin content in the fruiting body of *Inonotus hispidus*.

No.	Carbon Source	Nitrogen Source	pH	Solid-Liqui Ratio	Yield (%)
1	Maltose	Beef extract	Normal	1.1:1	31.8 ± 1.34
2	Maltose	Ammonium Tartrate	6	1.5:1	28.5 ± 0.86
3	Maltose	Tryptone	7	1.4:1	29.8 ± 0.59
4	Fructose	Beef extract	7	1.5:1	30.9 ± 0.57
5	Fructose	Ammonium Tartrate	Normal	1.4:1	29.8 ± 1.64
6	Fructose	Tryptone	6	1.1:1	27.8 ± 0.33
7	Mannitol	Beef extract	6	1.4:1	29.1 ± 0.25
8	Mannitol	Ammonium Tartrate	7	1.1:1	29.7 ± 0.52
9	Mannitol	Tryptone	Normal	1.5:1	29.2 ± 0.49
Mean 1	30.033	30.600	30.267	29.767	
Mean 2	29.500	29.333	28.467	29.533	
Mean 3	29.333	28.933	30.133	29.567	
Range	0.700	1.667	1.800	0.234	

**Table 6 foods-10-02893-t006:** Variance analysis of melanin content in the fruiting body of *Inonotus hispidus*.

Source	Sum of Squares	Df	Mean Square	F	Significance
Carbon source	0.802	2	0.401	0.280	<0.05 *
Nitrogen source	4.542	2	2.271	1.583	<0.05 *
Solid-liqui ratio	0.096	2	0.048	0.033	<0.05 *
pH	6.036	2	3.018	2.104	<0.05 *
Error	11.48	8			

* *p* < 0.05 means statistical significance.

## Data Availability

The data that support the findings of this study are available from the corresponding author, Junsheng Fu, upon reasonable request.

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
