# Peer review of "Optimization of Solid-State Fermentation Extraction of Inonotus hispidus Fruiting Body Melanin"

_foods, 2021, doi:10.3390/foods10122893_

Round 1
Reviewer 1 Report
this manuscript represents the new frontier on the use of extraction processes in order to improve yield. However, I believe it is necessary to carry out further investigations on the purpose of its use. The extraction phase and therefore the purpose of the manuscript is clear and well described but it is an antitumor part I believe requires further experiments.
Author Response
Respond: Thanks to the expert’s careful review, In this study, through the optimization of complex enzyme extraction and the screening of solid-state fermentation substrate conditions, a fermentation and extraction condition for high-yield Inonotus hispidus melanin was explored. The extraction rate of melanin could reach 33%. The structure of melanin was characterized by infrared absorption spectroscopy and electron microscopy scanning. This article explored the inhibitory effect of Inonotus hispidus melanin on the growth of different cancer cells.
Importantly, the core of this research is to explore the formulation and conditions of the solid-state fermentation medium for high melanin production. The MTT method is used to initially explore the inhibitory effect of melanin on cancer cells. The core of the article is not to explore the anti-cancer activity of melanin. The MTT in vitro cell proliferation assay is one of the most widely used assays for evaluating preliminary anticancer activity of both synthetic derivatives and natural products and natural product extracts (McCauley J., Zivanovic A., Skropeta D., Metabolomics Tools for Natural Product Discovery, 2013). The research team will further explore the anti-cancer activity of melanin in vivo in the future, so t this part will not be introduced too much in this article. The corresponding part of the article has been adjusted in language and marked in red. Thank you.
Reviewer 2 Report
The ‘Abstract’ is concise, specific.
The ‘Introduction’ is adequate to the topic.
The ‘Material & Methods’ is correct.
The ‘Results’ showed correct, according to ‘Methods’.
The ‘Discussion’ is properly prepared. The extraction rate, Authors achieved, was discussed with literature.
The ‘Conclusions’ are correct, however, I did not find any propositions about further studies or possibilities to broaden the analysis they showed in the manuscript.
References: corrections needed, as required by the journal
The manuscript requires some minor editorial corrections, however, this does not detract from the value of the work. The manuscript is written in a valid and scientific language that conforms to the requirements for similar articles. It brings important scientific value.
Line 6: Remove ‘Affiliation 1’ sentence
Throughout the text: add space between every number and unit ex. 20g -> 20 g
Line 39: please correct Reference [8] in text -> NE remove
Abbreviations need to be explained when appeared for the first time (ex. IHFM, etc.)
Line 57: remove space before brackets
Line 74: remove the duplicated bracket
Line 75: a period. in place of the semicolon ;
Line 122: ’75 Rinse with %’ ?? – correction needed
Line 123: What was centrifugation speed? 10,000 (ten) or 10000 (ten thousands) r/min ?
Line 177-178: the sentence needs correction
Line 467-469 – remove.
Author Response
Thank you reviewers for their comments. References have been corrected as required, and other parts of the article have been adjusted in language and format and marked in red. Thank you.
Reviewer 3 Report
- This study design has some problems. The most serious part is about anti-cancer. Cancer cell survival is not equal to anti-cancer.
- Anything is toxic, and the toxicity is based on the dosage. Please provide the toxicity of each fermented materials and then move to advanced anti-cancer study
Author Response
1.This study design has some problems. The most serious part is about anti-cancer. Cancer cell survival is not equal to anti-cancer.
Respond 1: Thank you for the reviewer’s comments. We highly agree with your point “cancer cell survival is not equal to anti-cancer”,
Therefore, we have made certain changes to the language expression in the article. This study only initially explored the inhibitory effect of IHFM on several different cancer cells through the MTT method. The MTT in vitro cell proliferation assay is one of the most widely used assays for evaluating preliminary anticancer activity of both synthetic derivatives and natural products and natural product extracts (McCauley J., Zivanovic A., Skropeta D., Metabolomics Tools for Natural Product Discovery, 2013; Thangakumar Arunachalam et al., Environmental Science and Pollution Research, 2020). Therefore, through the MTT method, this study has concluded that Inonotus hispidus melanin has a certain inhibition in cancer cells. The research team will conduct apoptosis and live animal enograft tumort experiments to further confirm the anticancer effect of melanin.
- Anything is toxic, and the toxicity is based on the dosage. Please provide the toxicity of each fermented materials and then move to advanced anti-cancer study
Respond 2: Thank the reviewers for their comments. In this study, the solid-state fermentation used rice medium, the substrate was rice, and the added nutrient solution was carbon and nitrogen sources commonly used in microbial experiments. They are all reagents and consumables commonly used in microbial experiments, such as glucose and peptone, and the dosage is within a reasonable range, however, the toxicity of each fermented material is not explored in this article. The melanin obtained by solid-state fermentation proved to be non-toxic to normal liver cell LO2 through MTT experiment, it shows that the treatments of various concentrations of melanin have no toxicity to normal cells, and it also indirectly proves that each fermented material has no obvious effect on melanin, but this part of the experimental data is being submitted to another journal.
Round 2
Reviewer 3 Report
- pH is important for fermentation. However, the pH control is so called natural pH. This description is amazing.
- Melanin is mainly about its protection on skin, especially UV light damage. Regarding the anti-cancer study design, it's quite rough. At least, normal cells could be involved.
- Survival of carcinoma is not equal to anti-cancer. Cell flow cytometry could be provided, and also the morphology is needed.
- What is the type for application in the future? Food type or ointment?
- The results are still very far from anti-cancer
Author Response
- pH is important for fermentation. However, the pH control is so called natural pH. This description is amazing.
Thanks to the expert's opinion, the natural pH mentioned in this article refers to the pH value of the medium which has not been adjusted. The statement may not be appropriate, the author has changed natural PH to normal PH.
In this paper, pH 5, 6, 7, 8, 9, and normal pH (pH=6.7) were set to cultivate Inonotus hispidus to extract melanin. The results showed that the extraction rate of melanin was the highest under normal pH(pH=6.7) culture conditions.
- Melanin is mainly about its protection on skin, especially UV light damage. Regarding the anti-cancer study design, it's quite rough. At least, normal cells could be involved.
Thanks for the comments of the experts. The IHFM anti-cancer research is only a little involved in this article. The author decided to study in depth and would publish it in another article. Therefore, the relevant content in the article will be deleted and no further explanation will be given. Thank you for your valuable advice.
- Survival of carcinoma is not equal to anti-cancer. Cell flow cytometry could be provided, and also the morphology is needed.
Thanks to the expert’s advice, We decided to delete the anti-cancer part of the article, subsequent experiments such as cell flow cytometry will be conducted later.
- What is the type for application in the future? Food type or ointment?
Thanks for the comments of the experts. As a natural active ingredient, melanin has wide medicinal value and biological activity, which can convert 99.9% of the energy from sunlight into thermal energy, effec-tively reducing the probability of skin cancer (Ye, Y.et al., Sci Immunol, 2017), besides, melanin also has whitening effects. Therefore, the laboratory has developed melanin masks, toothpastes, jellies and other products, which are currently in the testing phase.
- The results are still very far from anti-cancer
Thanks for the comments of the experts. This study only concluded that Inonotus hispidus melanin has a certain inhibition in cancer cells. The research team will conduct apoptosis and live animal enograft tumort experiments to further confirm the anticancer effect of melanin. Therefore, this article deletes the part related to the cell experiment for follow-up research.